# A Hybrid Model Consisting of Supervised and Unsupervised Learning for Landslide Susceptibility Mapping

**Zhu Liang [1], Changming Wang [1,*] , Zhijie Duan [2], Hailiang Liu [1], Xiaoyang Liu [1] and Kaleem Ullah Jan Khan [1]**

[1]  College of Construction Engineering, Jilin University, Changchun 130012, China;
liangzhu19@mails.jlu.edu.cn (Z.L.); hailiang19@mails.jlu.edu.cn (H.L.); liuxiaoy19@mails.jlu.edu.cn (X.L.);
Khan2417@mails.jlu.edu.cn (K.U.J.K.)

[2]  State Key Laboratory of Hydroscience and Engineering Tsinghua University, Beijing 100084, China;
duanzj17@mails.tsinghua.edu.cn

*  Correspondence: wangcm@jlu.edu.cn; Tel.: +86-135-0441-8751

**Abstract:** Landslides cause huge damage to social economy and human beings every year. Landslide susceptibility mapping (LSM) occupies an important position in land use and risk management. This study is to investigate a hybrid model which makes full use of the advantage of supervised learning model (SLM) and unsupervised learning model (ULM). Firstly, ten continuous variables were used to develop a ULM which consisted of factor analysis (FA) and k-means cluster for a preliminary landslide susceptibility map. Secondly, 351 landslides with "1" label were collected and the same number of non-landslide samples with "0" label were selected from the very low susceptibility area in the preliminary map, constituting a new priori condition for a SLM, and thirteen factors were used for the modeling of gradient boosting decision tree (GBDT) which represented for SLM. Finally, the performance of different models was verified using related indexes. The results showed that the performance of the pretreated GBDT model was improved with sensitivity, specificity, accuracy and the area under the curve (AUC) values of 88.60%, 92.59%, 90.60% and 0.976, respectively. It can be concluded that a pretreated model with strong robustness can be constructed by increasing the purity of samples.

**Keywords:** landslide susceptibility; unsupervised machine learning; supervised machine learning; hybrid model; geographic information system

## 1. Introduction

Landslides are one of the most destructive natural disasters worldwide resulting in inestimable loss of lives and economics [1]. Therefore, it is vital to distinguish whether an area is prone to landslides to avoid or decrease unnecessary losses. Landslide susceptibility mapping (LSM) divides the area into different levels based on the probability of landslides, and it is significant for land use and landslide prevention and mitigation [2].

To obtain a landslide susceptibility map mainly involves two steps: (1) Collect related data like conditioning factors and landslide information; (2) select the most suitable model. With the development of computer technology and related theories, the acquisition of data has become easier, and numerous methods have been applied to LSM. In general, various models can be divided into heuristic, physical, traditional statistical and new machine learning [3]. In recent years, the emergence of ensemble model has become popular due to their high accuracy and generalization ability [4,5].

Data-driven approach for LSM can be also split into unsupervised and supervised learning depending on whether prior conditions are need. The prior conditions refer to a number of samples with label assigned in advance, which are used for model training and verifying. The modeling of unsupervised learning only needs conditioning factors but prior conditions. Unsupervised learning model (ULM) commonly used in LSM includes factor analysis (FA), clustering analysis and principal component analysis [6,7]. The accuracy of

ULM is hard to verify and therefore unconvincing [8]. However, ULM is powerful in terms of dimensionality reduction, which is also important for data processing [9,10]. Supervised learning model (SLM) usually obtains a more accurate landslide susceptibility map [11,12]. Labeled samples acquire both negative and positive samples, which refer to non-landslide samples and landslide locations in LSM, respectively. With the development of GPS, remote sensing, etc., the determination and collection of landslide locations has become easier and more reliable, while the non-landslide samples are invisible and unpredictable and the selection becomes difficult or optional, which brings noise for data and finally leads to an unsatisfactory model. The SLM frequently used in LSM includes logistic regression, random forest, artificial neural network and support vector machine [13–15].

The selection of the most suitable models for LSM has been discussed for many years, and no consensus is reached because the performance varies from different study areas. Accordingly, comparison of different methods for LSM is necessary and some researches have been done [16–18]. In this study, we proposed a hybrid model with the full use of the advantage of ULM and SLM. Firstly, FA and k-means cluster were combined to obtain an initial landslide susceptibility map, and the accuracy was verified by calculating the information value with the existing landslide samples. Secondly, non-landslide samples were selected from the very low susceptibility area predicted by the ULM. A high-quality sample dataset consisting of landslide and non-landslide samples was used for the modeling of GBDT, an ensemble learning model for LSM. Then, the performance of the improved GBDT model was verified according to related indexes and compared to a normal GBDT model, the non-landslide samples of which were selected randomly. Finally, major conditioning factors were analyzed by FA and Gini index [19].

To summarize, the aim of this study is to explore a new hybrid model for LSM with high accuracy by selecting the non-landslide samples in a reliable way. The existing landslide samples were used as posterior condition for the ULM, and the results obtained by the ULM were used as a new prior condition to improve the purity of samples for SLM. Huairou district in Beijing was selected as the study area, where landslides occurred frequently. Finally, thirteen conditioning factors were prepared for modeling; LSM was created by different models, and the most suitable model was determined after evaluation and comparison.

## 2. Materials

### 2.1. Study Area and Landslide Inventories

Huairou district is located in the Northeast of Beijing, and it covers an areas of 2123 km$^2$ with a population of 422,000 (Figure 1). Rainfall, as a major triggering factor, is closely related to the occurrence of landslides in this study area. For example, continuous and heavy rainfall in 1969, 1972 and 1991 caused a sizable number of landslides. It has a warm temperate semi-humid climate with the annual rainfall ranging from 600–700 mm mainly concentrated in June to August.

Faults and folds are developed and mainly distributed in the middle of the study area (Figure 2). The lithology is relatively fractured, and the main kinds of lithology exposed in field investigation are mudstone, breccia and shale (Figure 2). Besides, granite from Jurassic (J), dolomite from CH and andesite from cretaceous also appear. Its elevation ranges from 23 m to 4413 m with slope angles varying from 0° to 87°.

The developed tourism resources in the research area attract a large number of tourists from all over the country every year. Natural disasters, especially landslides, have caused damage to infrastructure (roads and houses) and posed a threat to human activity.

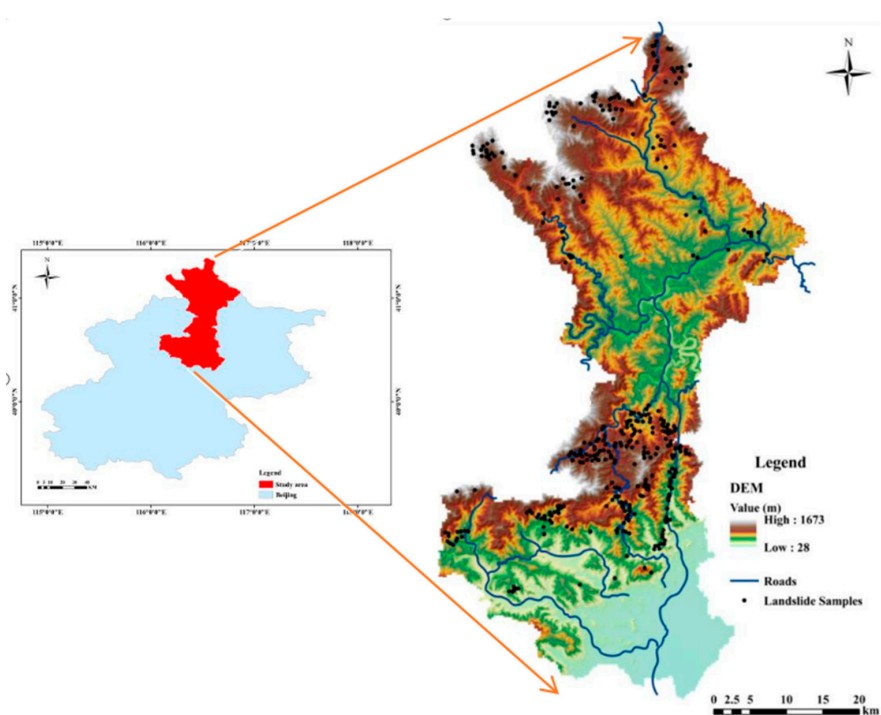

**Figure 1.** Map of research location showing landslide inventory.

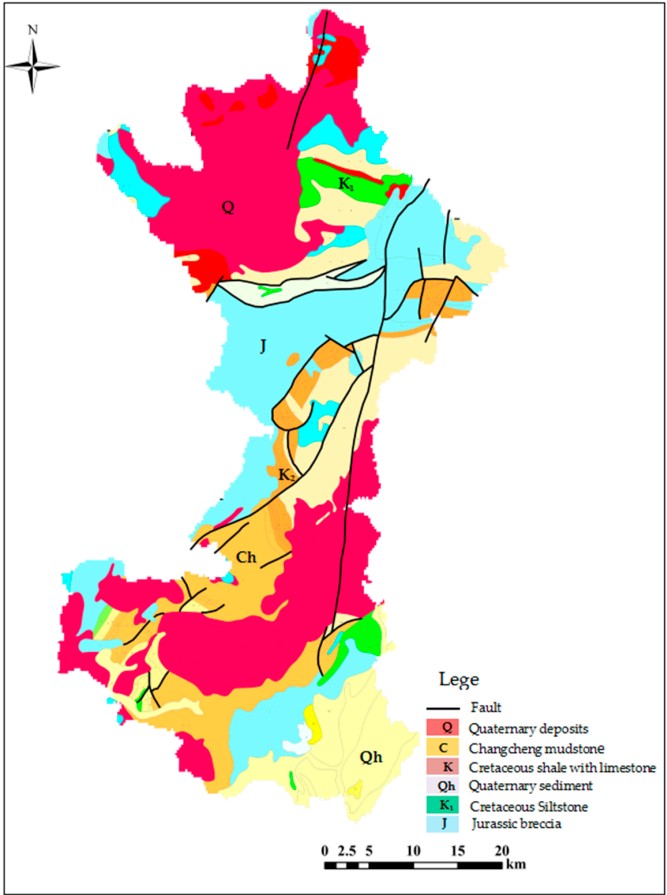

**Figure 2.** The geology map of study area.

As for supervised learning models, collecting landslide samples is a key part for model training, and therefore, a complete landslide inventory map is indispensable. It consists of 351 landslides depicted from reports (1970–2010), field surveys (from 2016–2017) (Figure 3) and Google Earth satellite images interpretation (May 2016). The area of the landslide we explored ranges from 0.05 to 1 km$^2$, and the average value reaches 0.26 km$^2$. All the landslide we considered in the study belongs to rainfall induced landslides which are fast and occurred suddenly.

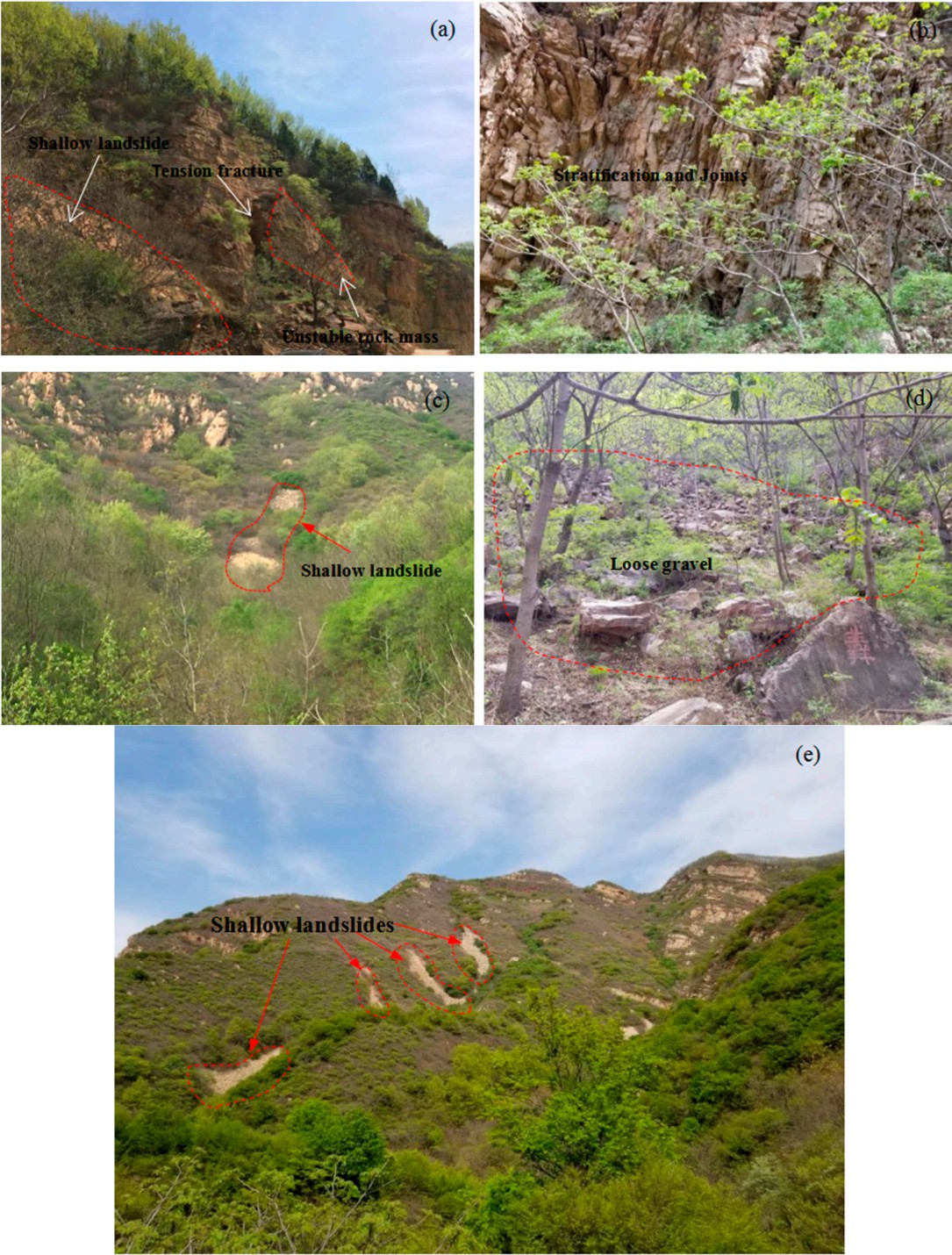

**Figure 3.** Field investigation photos. (**a**) Shallow landslide in Tang Hekou town; (**b**) stratification and joints in Tang Hekou town; (**c**) shallow landslide in Trumpet Gate township; (**d**) loose gravel distributed in Zheng fence village; (**e**) shallow landslides in the north bay village.

## 2.2. Data Preparation

### 2.2.1. Mapping Units

A suitable mapping unit should be determined before the modeling of LSM [20]. Grid cells and slope units are two of the most popular units, and the detailed discussion between them can be referred to another literature [21]. Slope units perform better in terms of maintaining the topographic conditions and distinguishing the locations of landslide. Accordingly, slope units were used in our work. ArcGIS was applied to divide the study area into 12,237 units based on the hydrological analysis tool, and we made boundary adjustment combined with remote sensing image.

### 2.2.2. Conditioning Factors

Conditioning factors are essential regardless of supervised or unsupervised learning models. Landslide occurrence is the result of intrinsic and extrinsic factors as topographical, geological and triggering factors [22]. On the other hand, data availability and reliability should be also considered. In this study, 13 conditioning factors generated from related data were selected based on the experience of previous researches.

Rainfall has been widely considered as an essential factor responsible for landslide occurring [23,24]. The occurrence of a landslide is affected by intensity and durability of rainfall. Maximum 24 h rainfall and maximum 7 days rainfall as representative for triggering factors are collected from the precipitation station nearby (1981–2010) and kriging interpolation method is applied to generate the distribution map. These two maps were divided into 5 classes with 30 mm interval (Figures 3b and 4a). The rainfall data comes from Beijing Hydrology Handbook (BHH).

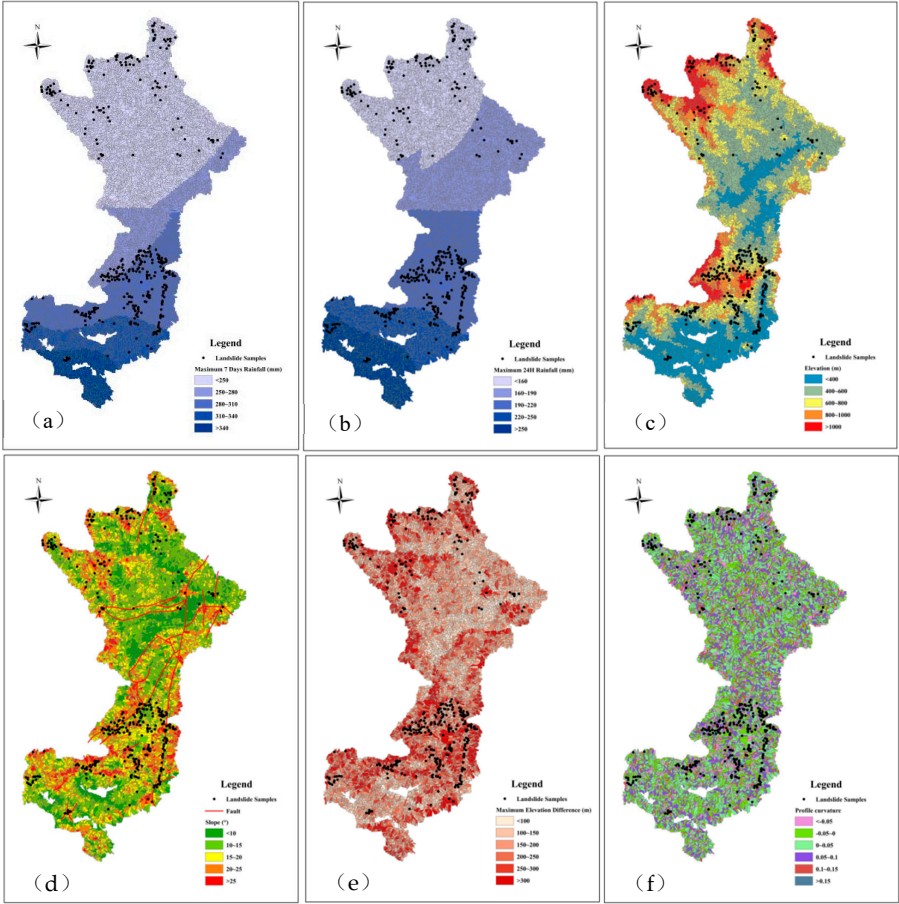

**Figure 4.** *Cont.*

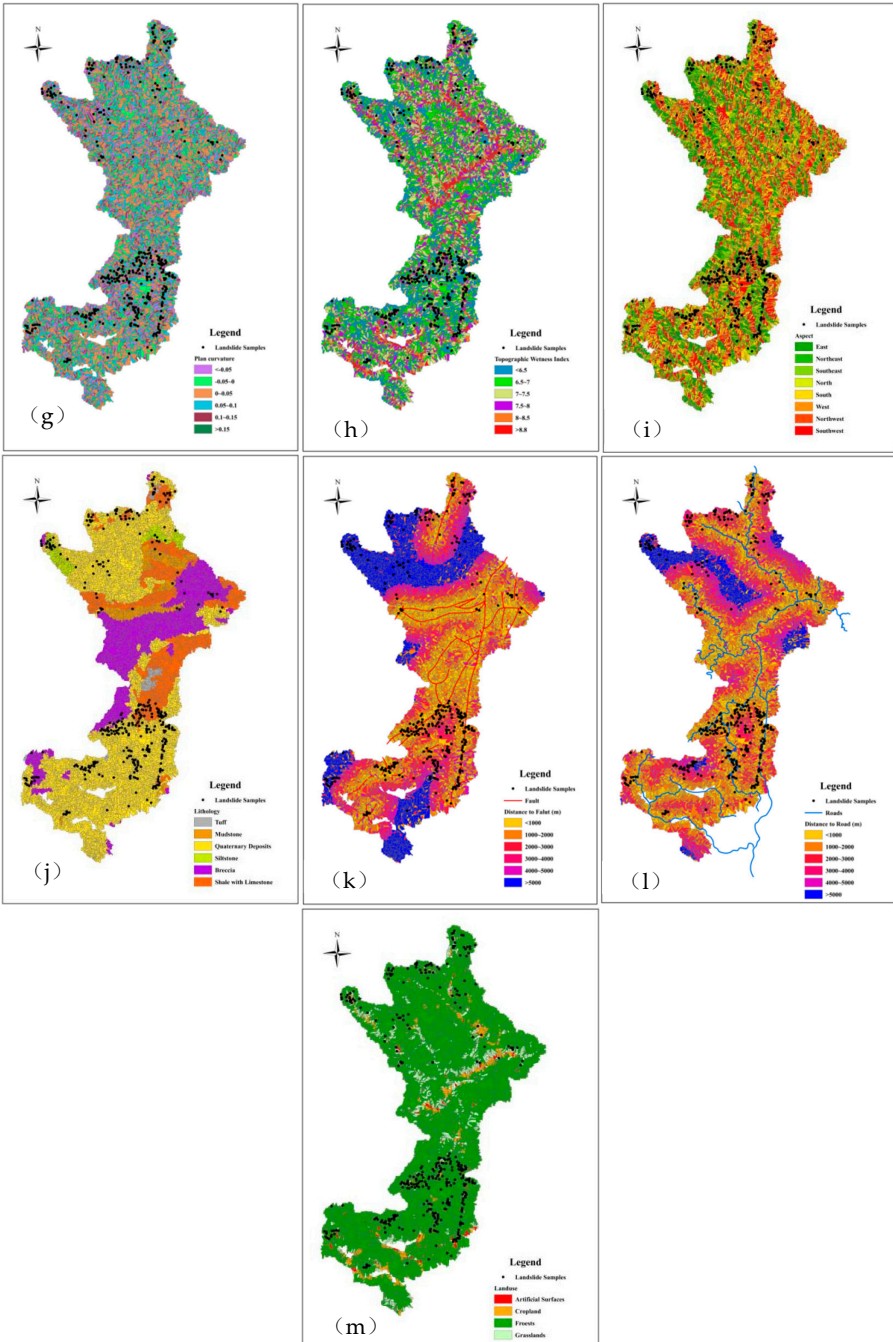

**Figure 4.** Thematic maps of different conditioning factors: (**a**) maximum 24h rainfall; (**b**) maximum 7 days rainfall; (**c**) elevation; (**d**) slope; (**e**) maximum elevation difference (MED); (**f**) profile curvature; (**g**) plan curvature; (**h**) topographic wetness index (TWI); (**i**) aspect; (**j**) lithology; (**k**) Distance to fault (DTF); (**l**) distance to road (DTR); (**m**) land use.

Seven topographic related factors are interpreted from the digital elevation model (DEM) with a resolution of 30 m. They include altitude, slope, aspect, maximum elevation difference (MED), plan curvature, profile curvature and topographic wetness index (TWI).

Elevation which was applied many times affects slope loading, rainfall and vegetation, and it was important for LSM [25]. The elevation of the study area ranges from 23 to 4413 m above sea level, and the thematic map was reclassified into 5 classes using 400 m as an interval (Figure 4c). Slope is another factor used commonly and steep slope contributes to landslide occurring more significantly [26]. Slope reflects the potential energy of landslides. The slope angle varies from 0° to 87°, which was divided into 5 classes with 5° interval

(Figure 4d). MED also reflects the kinetic condition and is obtained by calculating the difference between the maximum and minimum values of elevation in the same slope unit [7]. It was divided into 6 classes with 50 m interval (Figure 4e). Curvature can be positive or negative, which reflects the unevenness of the ground and affects the confluence capacity of the surface [27]. Profile curvature and plan curvature were divided into 6 classes with intervals of 0.05 each (Figure 4f,g). TWI is a hydrological variable that reflects both slope and soil moisture content; its definition can refer to another paper [28], and it was divided into 6 classes with intervals of 0.5 (Figure 4h). Slope aspect has an impact on sunlight duration, rainfall and humidity, which has been quoted by many researches [29,30], and it was divided into 8 categories in this study (Figure 4i).

Two geological related factors like the lithology and faults are obtained from a geologic map at scale of 1:50,000. The mechanical properties of different lithology are varied (like shear strength), which contribute different vulnerability to landslide. Lithology is considered as one of the most essential factors for LSM [31,32]. There are mainly five lithological categories as tuff, mudstone, siltstone, breccia and limestone distributed in this study area (Figure 4j). Faults destroy the integrity of the rock formation, and weak planes are easier to form in slopes. Landslides tend to occur in dense fault zones [33]. Distance to fault is calculated by the spatial distance analysis tool in ArcGIS and divided into 6 classes with intervals of 1000 m (Figure 4k).

Road construction is the main engineering activity in the study area. Landslides are more likely to distributed nearby the roads in the study area. The engineering works will aggravate the already existing susceptibility to landslides. Unreasonable excavation and reconstruction of land usually lead to landslides. Similarly, distance to road is obtained according to the data form the Department of Natural Resources of Beijing (DNRB) and divided into 6 classes with intervals of 1000 m (Figure 4l).

Inappropriate land use is another triggering factor, which is regularly utilized in LSM [34]. Similarity, it reflects the influence of human activities on natural environment as surface coverage and the integrity of rocks [35]. Four categories as forests, grasslands, cropland and artificial surfaces are distributed in the study area (Figure 4m).

Three factors, slope aspect, lithology and land use, belong to categorical variable and the others are continuous variable (Table 1). For unified presentation, the continuous variables were reclassified into 5 to 8 classes (Figure 4a–m).

**Table 1.** Conditioning factors used in this study.

| Category | Conditioning Factors | Type | Data Source | Values |
|---|---|---|---|---|
| Topographical | Altitude (m) | Continuous | DEM | 23–4413 |
| | Slope angle (°) | Continuous | DEM | 0–87 |
| | MED (m) | Continuous | DEM | 12–652 |
| | Plan curvature | Continuous | DEM | −0.51–0.64 |
| | Profile curvature | Continuous | DEM | −0.86–0.56 |
| | Aspect | Categorical | DEM | East; Northeast; North; West; Northwest; South; Southwest; Southeast |
| | TWI | Continuous | DEM | 5.13–17.94 |
| Geological | Distance to faults (km) | Continuous | Geological map | <1; 1–2; 2–3; 3–4; 4–5; >5 |
| | Lithology | Categorical | GESI | 0–2.5; 2.5–5; 5–7.5; 7.5–10; 10–12.5; 12.5–15; 15–17.5; >17.5 |
| Triggering factors | Maximum 24 h rainfall (mm) | Continuous | BHH | 148.02–304.36 |
| | Maximum 7 days rainfall (mm) | Continuous | BHH | 211.36–376.44 |
| | Distance to roads (km) | Continuous | DNRB | <1; 1–2; 2–3; 3–4; 4–5; >5 |
| | Land use | Categorical | DNRB | Artificial Surfaces; Cropland; Forests; Grasslands |

## 3. Methodology

### 3.1. FA

FA is a usual unsupervised learning method for extracting common factors from variable groups, which solve the problem of high dimension and exploring major factors in LSM. Two matrices are established to observe the data in FA, one for common factors and the other for special factors, which can be expressed as

$$X = AF + \beta \tag{1}$$

where $\beta$ represents the special factors; $F$ represents the common factors; $A$ is the factor-loading matrix and $X$ is the original data.

The basic procedures of FA consist of four steps [7]:

(1)    Test the fitness of applying FA.

Bartlett-test of sphericity and Kaiser–Meyer–Olkin measure (KMO) are two indexes used to evaluate the fitness of applying FA to variables in the study.

(2)    Extraction factor.

Principal component analysis (PCA) is applied to extract factors and the cumulative variance contribution rate is over 85% of which the first m factors are remained.

(3)    Orthogonal rotation.

Varimax is used for rotating matrix and makes the significance of each common factor clearer.

(4)    Calculating factor scores.

Factor score is calculated by Thomson regression method as follows:

$$S = WX \tag{2}$$

where $X$ represents the factor-loading matrix; $W$ is coefficient matrix and $S$ is the factor scores.

### 3.2. K-Means Clustering

K-means clustering is another unsupervised learning algorithm and widely used in various aspects due to its simplicity and efficiency [36]. It divides n samples into k clusters based on Euclidean distances.

The procedures of K-means clustering consists of three steps:

(1)    Determining the initial clustering centers;
(2)    Calculating the Euclidean distances between samples and the clustering center;
(3)    Retrieves the centers for each new cluster and iterates until it meets the following equation:

$$\frac{|u_{n+1} - u_n|}{u_{n+1}} \leq \varepsilon \tag{3}$$

where $u_{n+1}$ is the sum of squares of distances after the $n$th iteration; $\varepsilon$ is the precision.

In this study, five initial clustering centers were determined randomly, and the iterations were set to be 20. The factor score of each unit was used as data input for clustering analysis.

### 3.3. Sampling and Validation Strategy

SLM applied to LSM belongs to a binary classification problem, the dataset of which contains both landslide and non-landslide samples at the ration of 1:1 [37]. A total of 351 landslide locations (positive samples) are assigned to "1", and the same number of non-landslide samples are assigned to "0" which are randomly selected from the lowest susceptibility area in the initial landslide susceptibility map predicted by ULM.

Without a proper verification process, the performance of a model is not convincing [38]. The 5-fold cross validation was utilized in this study, where the dataset is randomly divided into 5 independent groups, one of which is for testing and the remaining for training [39] and cycles verification 5 times.

### 3.4. GBDT

GBDT is one of the most famous and successful ensemble-learning algorithms applied in Boosting family [40]. It models with decision tree (DT) as the basic classifier and Gradient Boosting as the training strategy. Each new DT is established to reduce the residual of the previous model in the direction of the gradient, and the final conclusion is to integrate all models. It performs well in terms of preventing overfitting and has strong generalization ability compared to the other machine learning methods [41]. The final equation is as follows:

$$f(x) = f_0(x) + \sum_{m=1}^{M} \sum_{j=1}^{J} \gamma_{jm} I(x \in R_{jm}) \tag{4}$$

where $M$ is the iterations; $J$ is the number of leaf nodes; $\gamma$ is the optimum fitting value of the negative gradient of the loss function in the leaf node region, $I$ is the indicator function and $R_{jm}$ is the corresponding leaf node region of the $t$th DT.

### 3.5. Information Value Model

Information value (IV) model is a binary statistical method which is used to determine the relationship between landslides and conditioning factors. It reflects the effect of condition factors on the occurrence of landslide based on the density of landslides [40]. However, in this study, IV is used to evaluate the performance of both unsupervised and supervised learning methods applied to LSM.

$$I_{(A_{i-j})} = I_n \frac{n_i / s_i}{n / s} \tag{5}$$

where $n_i$ represents the area of landslide of the $i$th level susceptibility; $s_i$ represents the total area of landslides in the study area; $n$ represents the area of the $i$th level susceptibility; $s$ represents the area of total units in the study area.

The IV can be positive or negative and a positive value indicates a stimulative effect on landslide occurring, the greater the value, the higher the possibility of landslides, and vice versa.

### 3.6. Model Performance

Three statistical indexes as sensitivity, specificity and accuracy and receiver operating characteristic curve (ROC) were applied to evaluate the performance of SLM while IV was for both unsupervised and supervised learning models [41]. The evaluation using the training data reflects the fitting ability of the model while the testing data for generalization. Related equations are as follows:

$$Accuracy = \frac{TP + TN}{TP + TN + FP + FN} \tag{6}$$

$$Sensitivity = \frac{TP}{TP + FN}$$
$$Specificity = \frac{TN}{FP + TN} \tag{7}$$

where $TP$ (true positive) represents the number of landslide samples classified correctly; $FP$ (false positive) represents the number of misclassified landslide samples; $TN$ (true negative) represents the number of non-landslide samples classified correctly; $FN$ (false negative) represents the number of misclassified non-landslide samples.

The area under the ROC cure (AUC) is a standard indicator to measure the quality of models, and it is split into five classes as poor, normal, good, pretty good and excellent, the value of which ranges from 0.5–0.6, 0.6–0.7, 0.7–0.8, 0.8–0.9, and 0.9–1, respectively.

The methodology applied in this study was showed in Figure 5, which is a flowchart of this work.

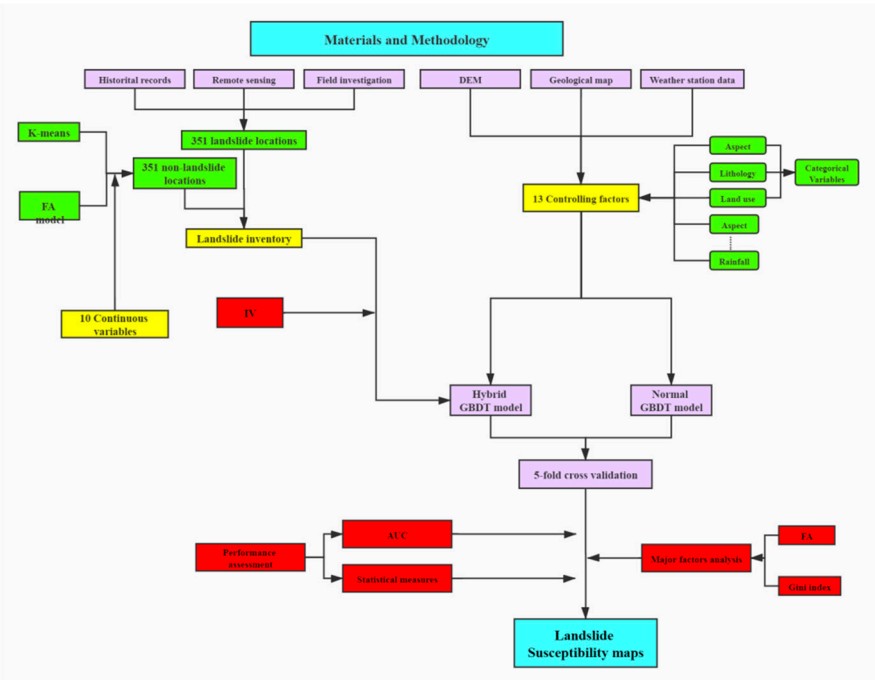

**Figure 5.** Flowchart of the proposed methodology in this study.

## 4. Results

### 4.1. LSM Obtained by the ULM

Ten continuous variables were input for the modeling of FA and the factors aspect; lithology and land use were not considered. The value of KMO was 0.765, which indicated a strong correlation between variables and was suitable for the modeling of FA. Table 2 showed that the accumulative contribution of the first five common factors (C1, C2, C3, C4 and C5) reached 86.501%, which retained most of the original information. The first two common factors accounted for a larger proportion as 28.993% and 22.842%, respectively. Based on the coefficients between C1 and conditioning factors, rainfall was regarded as the major factors for landslide occurrence. C2 highlighted the importance of slope and MED while C3 focused on the factor distance to fault and distance to road. C4 and C5 emphasized the importance of curvature.

**Table 2.** The coefficients between common factors and conditioning factors after Varimax rotation.

| Factor | C1 | C2 | C3 | C4 | C5 |
|---|---|---|---|---|---|
| Distance to fault (F1) | −0.032 | 0.027 | 0.895 | 0.021 | 0.005 |
| Plan curvature (F2) | −0.004 | −0.004 | −0.028 | −0.135 | 0.983 |
| Profile curvature (F3) | 0.002 | 0.007 | −0.032 | 0.940 | −0.091 |
| Distance to road (F4) | −0.177 | 0.067 | 0.831 | −0.070 | −0.030 |
| Slope (F5) | 0.024 | 0.909 | 0.046 | −0.168 | −0.049 |
| Elevation (F6) | −0.544 | 0.438 | 0.409 | −0.264 | −0.170 |
| MED (F7) | 0.050 | 0.875 | 0.048 | 0.069 | 0.017 |
| Maximum 7 days rainfall (F8) | 0.977 | 0.057 | −0.133 | −0.017 | −0.015 |
| Maximum 24H rainfall (F9) | 0.979 | 0.070 | −0.053 | −0.023 | −0.022 |
| TWI (F10) | 0.008 | −0.593 | −0.051 | 0.658 | −0.131 |
| Contribution rate (%) | 28.993 | 22.842 | 14.598 | 11.974 | 8.121 |
| Accumulative contribution (%) | 28.993 | 51.834 | 66.433 | 78.380 | 86.501 |

Accordingly, the scoring functions were determined as follows:

$$S = 0.13 \times F1 + 0.056 \times F2 - 0.05 \times F3 + 0.194 \times F4 + 0.215 \times F5 + 0.14 \times F6$$
$$+0.205 \times F7 + 0.075 \times F8 + 0.097 \times F9 - 0.175 \times F10 \tag{8}$$

Factor score reflects the contribution of conditioning factors to landslide occurring. The higher the score, the higher susceptibility of landslide. The score was a series of discrete values ranging from −3.50–3.29, and there was no obvious turning point for classification. In this study, the factor scores of each unit were classified by k-means cluster, and LSM was obtained according to five classes as very high, high, moderate, low, and very low.

Table 3 shows the percentage of each susceptibility class predicted by FA. FA classified the 34.55% of area as moderate class, which occupied the largest proportion. Low susceptibility occupied the smallest proportion as 3.25% and very low as 16.8%. High and very high class voted for 33.51% and 12.01%, respectively. Besides, the IVs of very low, low and moderate class were −0.84, −1.07 and −0.26, respectively. While the IVs of high and very high were 0.27 and 0.57, respectively.

**Table 3.** The information value of landslide susceptibility for different levels.

| Model | Class | Total Area (m$^2$) | Percentage of Area (%) | Landslide Area (m$^2$) | Percentage of Landslide Area (%) | IV |
|---|---|---|---|---|---|---|
| FA | Very low | 299,175,346 | 16.80 | 6,615,258 | 7.25 | −0.84 |
|  | Low | 58,195,125 | 3.25 | 1,023,304 | 1.12 | −1.07 |
|  | Moderate | 619,590,418 | 34.55 | 24,292,631 | 26.64 | −0.26 |
|  | High | 601,033,643 | 33.51 | 39,841,031 | 43.69 | 0.27 |
|  | Very high | 215,547,446 | 12.01 | 19,415,340 | 21.29 | 0.57 |
| FA+ GBDT | Very low | 360,203,145 | 20.08 | 2,802,398 | 3.08 | −1.87 |
|  | Low | 286,967,956 | 16.00 | 2,640,178 | 2.9 | −1.71 |
|  | Moderate | 173,449,271 | 9.67 | 4,451,170 | 4.9 | −0.68 |
|  | High | 325,321,468 | 18.78 | 10,752,965 | 19.38 | 0.03 |
|  | Very high | 647,600,138 | 35.46 | 73,787,162 | 69.72 | 0.68 |

The very low area was accounted for only 7.25% of landslide samples and 16.8% of the whole study area. Therefore, the LSM obtained from FA was regarded as the new prior conditions for supervised learning, and 351 non-landslides samples were determined from the area with very-low susceptibility in the landslide susceptibility map predicted by FA.

*4.2. LSM Obtained by GBDT*

The data consisting of 702 samples (351 landslide samples and 351 non-landslide samples) and thirteen conditioning factors were utilized as the inputs for GBDT modeling. To highlight the advancement of the improved model, another GBDT model which selected non-landslide samples randomly form the "landslide free area" was also constructed for LSM. As for training process, the improved model performed better with the sensitivity, specificity, accuracy and AUC values of 92.29%, 90.52%, 91.69% and 0.986, while GBDT model also performed well the sensitivity, specificity, accuracy and AUC values of 88.51%, 90.24%, 89.38% and 0.963 (Figure 6). Besides, the improved model outperformed again in validation (sensitivity = 88.60%, specificity = 92.59%, accuracy = 90.60% and AUC = 0.976). Similarly, the performance of GBDT model was satisfactory (sensitivity = 83.73%, specificity = 85.47%, accuracy = 84.62% and AUC = 0.937). The performance of the GBDT model in the validation data (accuracy = 84.62%) had dropped significantly compared to the training data (accuracy = 89.38%), and it manifested that the normal model was over-fitting and thus indicated a limited ability of generalization (Table 4). Therefore, the improved model combining with FA and GBDT was effective and was applied to calculate the landslide susceptibility index (LSI) for the whole study area. Similarly, the study area was reclassified into five classes based on LSI (Table 3). Of the study area, 35.46% was categorized as very high susceptibility, and it occupied the greatest proportion, followed by high susceptibility as 18.78%. The very low susceptibility accounted for 20.08% of area and low susceptibility for 16.0%. As for moderate level, it occupied 9.67% of the area, which was the smallest

proportion. The IVs of high and very high level were both positive as 0.03 and 0.68, respectively, while the IVs of very low, low and moderate susceptibility were all negative as −1.87, −1.71 and −0.68, respectively.

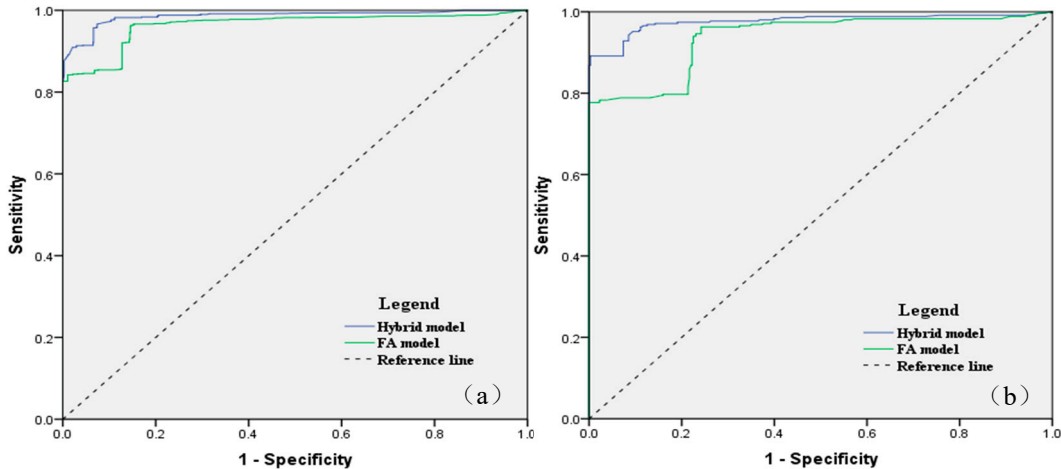

**Figure 6.** ROC curves of the SLM: (**a**) ROC cure using training dataset; (**b**) ROC cure using validation dataset.

**Table 4.** Performance of two models.

| Dataset | Metrics | Normal GBDT | Hybrid Model |
|---|---|---|---|
| Training | Sensitivity | 88.51% | 92.29% |
| | Specificity | 90.24% | 90.52% |
| | Accuracy | 89.38% | 91.69% |
| | AUC | 0.963 | 0.986 |
| Test | Sensitivity | 83.73% | 88.60% |
| | Specificity | 85.47% | 92.59% |
| | Accuracy | 84.62% | 90.60% |
| | AUC | 0.937 | 0.976 |

Gini index was applied in SLM to analyze the major conditioning factors for LSM. The larger the Gini index indicated, the greater the contribution of conditioning factors to landslide occurring. Ten conditioning factors with importance remained and were normalized as shown in Table 5. Slope, MED, TWI and elevation were regarded as the major factors with the weigh values of 0.25, 0.23, 0.18 and 0.1, respectively. The weigh values of categorical variables as lithology and land use were both 0.06, which were also important. The factors with lower weigh values as DTR, rainfall and profile curvature had limited effect on the occurrence of landslide.

**Table 5.** Variables importance assigned by Gini index.

| Method | Slope | MED | TWI | Elevation | Lithology | land Use | DTR | Maximum 24 h Rainfall | Profile Curvature | Maximum 7 Days Rainfall |
|---|---|---|---|---|---|---|---|---|---|---|
| Gini index | 0.26 | 0.24 | 0.19 | 0.1 | 0.06 | 0.06 | 0.03 | 0.02 | 0.02 | 0.02 |

*4.3. Comparison of Different Models for LSM*

4.3.1. Selection of the Major Conditioning Factors

Table 6 showed the analysis results of conditioning factors by FA and Gini index. To highlight the difference between the FA and Gini index, we rearranged the conditioning factors according to the following rules: The most important factor is defined as "1", followed by "2", and so on. Slope and MED played an important role in both models. Rainfall ranked the 1st in FA model while the last in Gini index. DTR ranked the fourth in

FA model while the last in Gini index. FA highlighted the significance of DTF while Gini index focused on TWI and elevation. Two categorical variables as lithology and land use appeared in Gini index (Figure 7 and Table 6).

**Table 6.** Comparison of major factors assigned by FA and Gini index.

| Factor<br>Method | Rainfall | Slope | MED | DTR | DTF | Curvature | TWI | Elevation | Lithology | Land Use |
|---|---|---|---|---|---|---|---|---|---|---|
| FA | 1 | 2 | 3 | 4 | 5 | 6 | | | | |
| Gini index | 7 | 1 | 2 | 7 | | 7 | 3 | 4 | 5 | 6 |

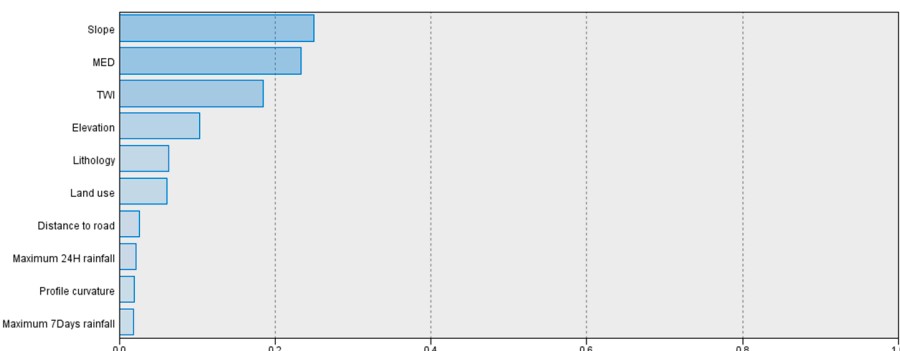

**Figure 7.** Relative importance of conditioning factors predicted by Gini index.

Rainfall as the only triggering factor used in this study deserves more attention because landslides preferred to occur with heavy or continuous rainfall according to historical records. Slope and MED reflect the dynamic conditions of landslides, which have been applied in many researches [3,41]. Engineering works (roads) adversely affect the stability of earth and rock massifs, and landslide locations are usually distributed along roads [42,43]. Accordingly, DTR should be considered as the major factor, and adequate adaptation engineering measures in the neighboring massifs of earth and rock should be designed. Similarly, DTF accounts for an indispensable position. Therefore, the major conditioning factors assigned by FA were more credible and receivable.

### 4.3.2. Accuracy and Rationality of LSM

IV was used as a standard to assess the performance of FA and the improved model for LSM in this study. The performance of two models were both satisfactory as the low susceptibility areas accounted for a negative IV while the high for a positive IV. The distribution of landslide susceptibility area should meet two rules: (1) the determined landslide locations should be predicted in the very high susceptibility as many as possible; (2) moderate susceptibility area should occupy a smaller proportion while low susceptibility a larger proportion [44,45]. Compared to FA model, the landslide susceptibility map outputted by the improved model was more reasonable as (1) more landslide locations were predicted in the very high susceptibility area; (2) moderate susceptibility area occupied less proportion; (3) low susceptibility area occupied larger proportion (Figures 8 and 9).

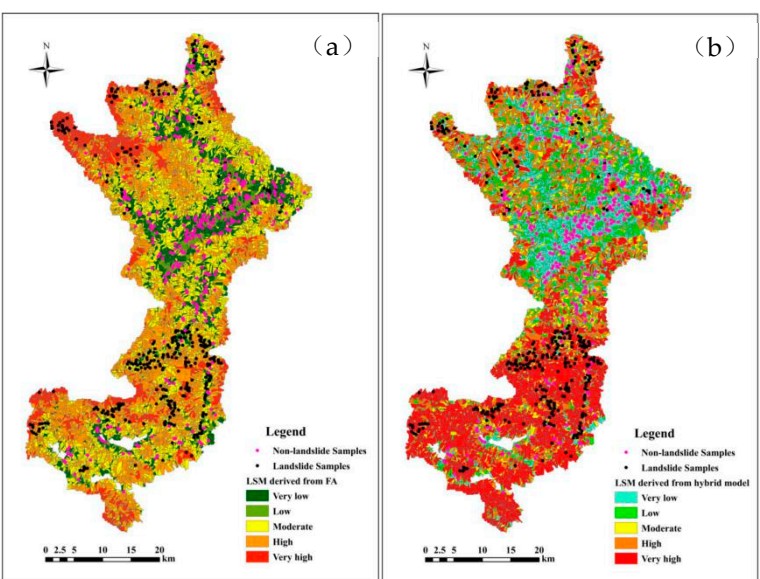

**Figure 8.** LSM obtained from the FA and hybrid model. (**a**) LSM obtained from the FA model, (**b**) LSM obtained from the hybrid model.

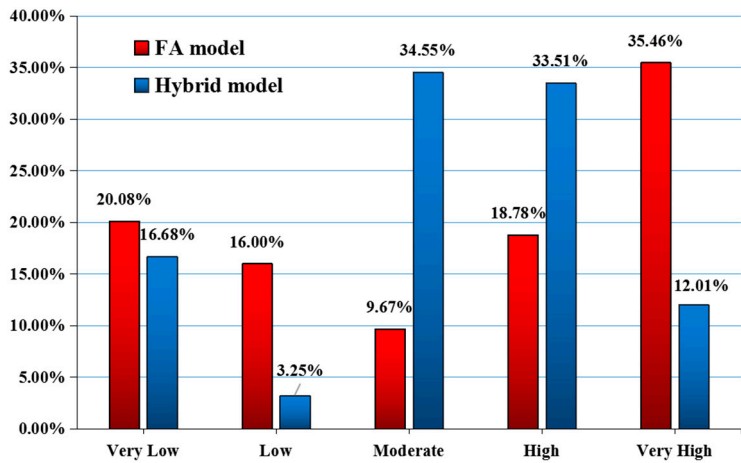

**Figure 9.** The percentage of the different susceptibility classes for FA and hybrid model.

Sensitivity, specificity, accuracy and the value of AUC were combined to assess the performance of the SLM for LSM. The improved model combining with FA and GBDT outperformed the GBDT model no matter in training or validation data. Therefore, the improved model was determined to be the most suitable model for LSM in this study.

## 5. Discussion

### 5.1. Comparison of Unsupervised and Supervised Learning for LSM

Compared to ULM, SLM is considered as a more suitable way for LSM as it performs better in terms of accuracy [46]. This is not only due to the rational use of label samples but also because the development of SLM presents more possibilities and advancements [47]. The ultimate goal of ULM is clustering rather than classification. Therefore, the application of ULM is more troublesome because the results obtained require further qualitative analysis. Previously, some studies performed further qualitative analysis on cluster samples based on the distribution density of landslides because clustering analysis lacks a standard to judge the landslide susceptibility of a sample after clustering [48]. However, labeled samples will no longer be suitable for verifying the accuracy of LSM predicted by unsupervised learning. In this study, the factor scores calculated by FA were regarded as

the standard for landslide susceptibility first and the results were then clustered. Finally, IV was used to verify the accuracy of LSM. In other words, labeled samples which are recorded landslide locations in landslide susceptibility modeling are used as posterior condition while factor scores as prior condition and cluster analysis as a tool of breakpoint like Natural breakpoint.

FA or principal component analysis are more often used to find out the major conditioning factors in LSM [49,50]. "Dimension Trouble" can be alleviated to some extent through omitting the factors with limited weight. However, the lower limit of the weight should be determined subjectively, and therefore, uncertainty is inevitable. In this study, the construction of common factors solved the problem of high dimension more effectively and scientifically. On the other hand, the major conditioning factors analyzed by Varimax rotation were more reasonable.

In spite of this, the results obtained by ULM were not reliable because no training and cross-validation steps are involved in the modeling process. SLM performed better in terms of accuracy due to having a determined target. The application of machine learning or deep learning has verified their advancement [51,52]. However, high accuracy plays the most important role in LSM but should not be the only consideration. The analysis of major conditioning factors is also of great interest, which is difficult to implement by machine learning methods for their "black box" training. Therefore, an improved model should not only focus on higher accuracy but on more reasonable analyticity.

### 5.2. Further Use of Prior Conditions

The premise of taking full advantage of SLM is to ensure the purity and quantity of samples. In other words, supervised learning requires high-quality samples, which also leads to the limited applicability for LSM. With the development of remote sensing imagery and UAV technology, the location of landslides has been more guaranteed. Landslide samples as the most important prior condition have received widespread attention. However, the collection of non-landslide samples becomes confused because they are unforeseeable and unreasonable selection will bring noise to the samples. Some researches select non-landslide samples randomly in the area where they consider it is "safe" based on some rules [53,54]. Besides, some studies select the samples based on k-means cluster [55,56]. In this study, non-landslide samples were selected in the very low susceptibility area predicted by the ULM. Labeled landslide samples were used to verify the performance of unsupervised learning method and then took the result as a new prior condition for SLM. Therefore, an improved supervised learning method was obtained, and it proved to be better than the normal SLM without new prior condition. The improved model used in this study solved the problem of over-fitting and improved the generalization ability. Besides, the distribution of landslide susceptibility area predicted by the improved model was more reasonable than the unsupervised learning method.

In previous researches, binary statistical method like frequency ratio (FR) was applied to determine the relationship between conditioning factors and landslide occurrence, and the FR values were used as the input for unsupervised learning method. Categorical variables are converted to a continuous variable trying to improve the performance of unsupervised learning method. However, FR fails to determine the relative importance of different conditioning factors. More importantly, the samples with label are more valuable and efficient for the modeling of supervised learning. In this study, FA was applied to explore the relative importance of conditioning factors and the labeled samples for the modeling of the improved model. Accordingly, the hybrid model can achieve great performance on condition that ensuring quality of samples and reasonable explanation of major conditioning factors should depend on FA.

## 6. Conclusions

LSM as an essential step in landslide prevention and mitigation has been developed for years, and the accuracy has been improved to varying degrees due to the development of

algorithms and technology. In the present study, a hybrid model consisting of unsupervised and supervised learning method was applied and compared in LSM, and the following conclusions can be drawn:

1.  FA performs well in dimensionality reduction and major conditioning factors analysis. Rainfall, slope, MED and DTR were considered as the major conditioning factors;
2.  The performance of the GBDT mode can be improved in terms of accuracy and generalization ability for the conditions that the quality of samples are guaranteed. The non-landslide samples selected from the very low susceptibility area predicted by the verified FA model were effective;
3.  The full utilization of prior conditions enhances the logicality of the models. Labeled samples were valuable in the validation of ULM and modeling of SLM;
4.  A hybrid model is recommended due to its high accuracy and reasonable explanation of major conditioning factors.

However, there are also some limitations in the study:

1.  More advanced methods need to be discussed and compared;
2.  The effect of other factors like mapping unit and interpretation accuracy of DEM was not considered;
3.  The hybrid model is not applied to other study area.

**Author Contributions:** Z.L., writing—original draft, methodology and software; C.W., validation; Z.D., code inspection; H.L. and X.L., software and validation; K.U.J.K., reviewing and editing. All authors have read and agreed to the published version of the manuscript.

**Funding:** This work was founded by Graduate Innovation Fund of Jilin University (101832020CX232) and the National Natural Science Foundation of China (Grant No. 41972267, 41977221, and 41572257).

**Data Availability Statement:** Python 3.7 using the GBDT class library of scikit-learn is responsible for the modeling of GBDT, the code of which is available at https://github.com/Liangzhu-mz/GBDT/blob/main/LSM (accessed on 9 April 2021). IBM SPSS Statistics Campus Edition is responsible for the modeling of Factor analysis and k-means clustering. ArcGIS 10.2 platform is responsible for hydrologic analysis, interpolation analysis and distance calculation and mapping the thematic maps. The code will work on workstations with at least 32GB of available memory and an NVIDIA GPU (NVIDIA 1080Ti or larger memory) to accelerate learning and prediction.

**Conflicts of Interest:** The authors declare no conflict of interest.

## Abbreviations

| | |
|---|---|
| LSM | Landslide susceptibility mapping |
| GBDT | Gradient boosting decision tree |
| SLM | Supervised learning model |
| ULM | Unsupervised learning model |
| ROC | Receiver operating characteristic curve |
| AUC | Area under the curve |
| FA | Factor analysis |
| DT | Decision tree |
| BHH | Beijing Hydrology Handbook |
| DEM | Digital elevation model |
| DNRB | Department of Natural Resources of Beijing |
| IV | Information value |
| TWI | Topographic wetness index |
| MED | Maximum elevation difference |
| DTR | Distance to road |
| DTF | Distance to fault |
| *TP* | True positive |
| *TN* | True negative |
| *FN* | False negative |
| *FP* | False positive |

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
