# Peer review of "A Hybrid Model Consisting of Supervised and Unsupervised Learning for Landslide Susceptibility Mapping"

_remotesensing, doi:10.3390/rs13081464_

Round 1
Reviewer 1 Report
Dear authors,
Please, read the attached file. Some clarifications and improvements are necessary.
Your reviewer
Author Response
Dear Revier
Thank you for your kindly work on our manuscript "A hybrid model consisting of supervised and unsupervised learning for landslide susceptibility mapping". The comment and suggestion you made were valuable and helpful for improving our work. Revised portion are marked in red in the paper. The main corrections in the paper and the responds to the Reviewer’s comments are as flowing:
- There is a need for clarification on the size (minimum considered area, as well as maximum landslide area) and the characteristics of the landslides: slow, fast landslides, etc.
Respond:Yes, we agree with the comment and have added related information of the landslide on page 3, line 107-110 .
- The landslides favored by the presence of roads are human induced landslides and should be treated distinctly. They cannot be merged with natural landslides (due to rainfall, slope, elevation etc). My suggestion is to treat distinctively the two kinds of landslides (natural and human induced). That means that you should delete the DTR as a major conditioning factor and to treat separately these landslides. If the susceptibility to landslides is determined before and after the construction of a road, the results will be different.
Respond:The comment is interesting.The inducing factors usually applied in LSM consists of human and natural factors. Rainfall as the major inducing factor have been applied for many times while human factor should not be ignored. Road construction is the main engineering project especially in the mountain area. According, the factor DTR is considered as the human induced factor and have also been applied for some researches. It needs more information of the time when the road was built and divided the landslide inventory separately. We would like to add more human inducing landslide factors and compare the results in detail in the following research and the ideal you provided is valuable.
- If the engineering works (roads) adversely affect the stability of earth and rock massifs, it would be useful to provide engineers the map with areas susceptible to natural landslides. The engineering works will aggravate the already existing susceptibility to landslides. In these areas, adequate adaptation engineering measures in the neighboring massifs of earth and rock should be designed (drainage systems, but not only). To what extent could the prediction for natural susceptibility be used by road engineers?
Respond:The comment is creative and worthy of further discussion. Most of the researches focus on the accuracy while as your question analyzed To what extent could the prediction for natural susceptibility be used by road engineers? If the human factor have the obvious influence on the susceptibility of landslide, we should pay special attention to areas that are of high susceptibility. However, the main aim of our study is to explore a new sampling way for LSM. We will add the related information on page 9, line 338-341.
- A suggestion for models validation would be to consider a number of “1” labelled samples as “0” labeled by error and to check the predicted susceptibility for landslides in these zones. 4. Page 4, Relation (1) is unreadable. Row 135 Probably, “siltstone” instead of “slitstone” Rows 158 and 183 In relation (1) ε represents the special factors, while in relation (3) is the precision. Please, use different notations. Who is R in relation (2) ? Relation (4)Relation (4) 1I suppose the index for the current leaf node should be “j” not “J”, where j=1,Who is I in the same relation ? Relation (5) I suppose In should bewritten In ; What is the difference between and I() ? Relation (6) The last term on the first line of relation (6) should be "0.14 x " instead of "0.14 x 6
Respond:The models validation has been fixed according to related indexes like AUC, accuracy, sensitivity and kappa. We've basically maintained a balance in the number of positive and negative samples to avoid bias. Accordingly, the results we obtained were reliable. On the other hand, it broadens our mind that we could change the labeled by error artificially and to check the predicted susceptibility for landslides in these zones. Of course, it needs further examination before we draw a reliable conclusion because we need to try different combination of the samples.
We are sorry for the mistakes to the equations and made the correction.
Thank you again for your kindly work.
Reviewer 2 Report
The aim of the paper entitled A hybrid model consisting of supervised and unsupervised learning for landslide susceptibility mapping is to explore a new hybrid model for Landslide susceptibility mapping with high accuracy by selecting the non-landslide samples in a reliable way. Overall paper is interesting and well prepared. Some aspects need improvement. Please explain all shortcuts not only in the abstract. Please add some information about the study area (geomorphology etc.). Please describe why these factors were used. Please add some table summarized used factors. Please describe used methodology of machine learning in particular. Why only AUC were used?
Author Response
Dear Reviewer
Thank you for your kindly work on our manuscript "A hybrid model consisting of supervised and unsupervised learning for landslide susceptibility mapping". The comment and suggestion you made were valuable and helpful for improving our work. Revised portion are marked in red in the paper. The main corrections in the paper and the responds to the Reviewer’s comments are as flowing:
1.The aim of the paper entitled A hybrid model consisting of supervised and unsupervised learning for landslide susceptibility mapping is to explore a new hybrid model for Landslide susceptibility mapping with high accuracy by selecting the non-landslide samples in a reliable way. Overall paper is interesting and well prepared. Some aspects need improvement.
Please explain all shortcuts not only in the abstract.
Respond:Thank you for your approval of our work. We try to improve the pure of the samples by combining different methods. It helps to further improve the accuracy of the models and provided a new way. We provide a Abbreviations table on page 1, line 29-39.
2.Please add some information about the study area (geomorphology etc.).
Respond:We have added some related information of the landslide on page 3, line 107-110 and provided the geology map (Figure 2).
3.Please describe why these factors were used.
Respond: Data availability, reliability and relevant experience from previous studies should be considered meanwhile. We have added some information concerning the factors on page 4.
4.Please add some table summarized used factors.
Respond: We have added the Table 1 on page 16, line 629.
Please describe used methodology of machine learning in particular.
Respond:We have added more information concerning the methodology on page 6, line 221-227.
5.Why only AUC were used?
Respond:The valued of AUC has been applied for many times considering its importance. We added extra indexes like accuracy, sensitivity and specificity to evaluate the performance of the models.
Thank you again for your kindly work.
Reviewer 3 Report
The manuscript was not formatted to remote sensing journal and difficult to follow the results.
I annotated the attached PDF for the authors. Please apply my comments properly.

Author Response
Dear Reviewer
Thank you for your kindly work on our manuscript "A hybrid model consisting of supervised and unsupervised learning for landslide susceptibility mapping". The comment and suggestion you made were valuable and helpful for improving our work. Revised portion are marked in red in the paper. The main corrections in the paper and the responds to the Reviewer’s comments are as flowing:
1.The format of the manuscript
Respond: We have removed the extra space.
2.The resolution of the figures
Respond: We provide a compressed package of the figures which are in high resolution.
3.The geology map
Respond: We have provided the geology map while it is not that clear (Figure 2).
- Some related information of lithology should be added.
Respond: We have provided the geology map (Figure 2) and related information of lithology on page 3, line 96-99.
5.What do you mean? I did not see any google earth image in your manuscript?
Respond: We applied the google earth image to help our field investigation. However, the google earth image is not available in China now and we can not provide the google earth images.
Round 2
Reviewer 1 Report
I agree with your comments and as I have understood you will consider some of my recommendations for future researches. There is only one correction which you omitted. The relation (4) should be written as in the attached file.

Author Response
Dear reviewer
Thank you again for your kindly work. Your comments have inspired our ideas and encouraged us to move forward.
Yours,
Zhu
Reviewer 3 Report
Accept in present form
Author Response
Dear reviewer
It is our pleasure to discuss with you and thank you again for your kindly work.
Yours,
Zhu